# Comparative efficacy and complication rates after local treatment for cervical intraepithelial neoplasia and stage 1a1 cervical cancer: protocol for a systematic review and network meta-analysis from the CIRCLE Group

Antonios Athanasiou,[1,2] Areti Angeliki Veroniki,[1,3] Orestis Efthimiou,[4] Ilkka Kalliala,[1,5] Huseyin Naci,[6] Sarah Bowden,[1,2] Maria Paraskevaidi,[1] Pierre Martin-Hirsch,[7] Philip Bennett,[1,2] Evangelos Paraskevaidis,[2,8] Georgia Salanti,[4] Maria Kyrgiou[1,2]

**Correspondence to**
Dr Maria Kyrgiou;
m.kyrgiou@imperial.ac.uk

## ABSTRACT

**Introduction** Local treatments for cervical intraepithelial neoplasia (CIN) and microinvasive disease remove or ablate a cone-shaped part of the uterine cervix containing the abnormal cells. A trend toward less radical techniques has raised concerns that this may adversely impact the rates of precancerous and cancerous recurrence. However, there has been no strong evidence to support such claims. We hereby describe a protocol of a systematic review and network meta-analysis that will update the evidence and compare all relevant treatments in terms of efficacy and complications.

**Methods and analysis** Literature searches in electronic databases (CENTRAL, MEDLINE, EMBASE) or trial registries will identify published and unpublished randomised controlled trials (RCTs) and cohort studies comparing the efficacy and complications among different excisional and ablative techniques. The excisional techniques include cold knife, laser or Fischer cone, large loop or needle excision of the transformation zone and the ablative radical point diathermy, cryotherapy, cold coagulation or laser ablation. The primary outcome will be residual/recurrent disease defined as abnormal histology or cytology of any grade, while secondary outcomes will include treatment failure rates defined as high-grade histology or cytology, histologically confirmed CIN1+ or histologically confirmed CIN2+, human papillomavirus positivity rates, involved margins rates, bleeding and cervical stenosis rates. We will assess the risk of bias in RCTs and observational studies using tools developed by the Cochrane Collaboration. Two authors will independently assess study eligibility, abstract the data and assess the risk of bias. Random-effects meta-analyses and network meta-analyses will be conducted using the OR for dichotomous outcomes and the mean difference for continuous outcomes. The quality of the evidence for the primary outcome will be assessed using the CINeMA (Confidence In Network Meta-Analysis) tool.

**Ethics and dissemination** Ethical approval is not required. We will disseminate findings to clinicians, policy-makers, patients and the public.

**PROSPERO registration number** CRD42018115508.

## Strengths and limitations of this study

► We plan to conduct the first network meta-analysis to assess the relative efficacy and complication rates of treatment methods for cervical preinvasive and early microinvasive disease.

► This study will produce comprehensive summaries of the clinical ranking of treatments and will employ methodologies that will allow the use of both randomised and observational data, aiming to use all published evidence.

► The results will inform clinicians, patients and clinical guidelines and will allow effective patient counselling at colposcopy clinics.

► We expect to find retrospective observational studies at high risk of recall, selection and publication bias. We will try to overcome this limitation by employing methods that aim to minimise bias.

## INTRODUCTION

Organised screening programmes in countries such as the UK have led to a dramatic decrease in the incidence and mortality from cervical cancer, especially when compared with the corresponding statistics for the other major cancers. Over a 20-year period, from 1993 to 1995 to 2013–2015, the overall age-standardised incidence rate of cancer in females increased by 16% in the UK,[1] whereas the corresponding data for cervical cancer showed a decrease of 24%.[2] Cervical cancer is largely preventable through detection and treatment of the preinvasive precursor, cervical intraepithelial neoplasia (CIN).[3] The local treatment methods are divided into excisional and destructive (ablative) that aim to remove or ablate, respectively, a

cone-shaped part of the cervix that contains the 'transformation zone' with the precancerous cells. Although large loop excision of the transformation zone (LLETZ) is the most commonly used methods in the UK[4] given its ease of execution and low cost, the preference of techniques varies across Europe and internationally.

A Cochrane systematic review of randomised controlled trials (RCTs) concluded that all local treatment techniques are highly effective in preventing preinvasive recurrence.[5] However, this review found no evidence of difference in treatment failure rates among the treatment techniques. This could be because the RCTs and the subsequent meta-analysis might have been underpowered to detect differences between the treatments. The largest study recruited only 390 participants,[6] while the majority of the rest were much smaller. A larger population-based study from Sweden,[7] which included 150 883 women diagnosed and treated for CIN3 (3 148 222 woman-years), reported a doubled standardised incidence ratio for post-treatment invasive recurrence during the follow-up period of around four decades in comparison to the general population, and initiated debates on the impact that less radical treatments may have on the subsequent risk of invasion.[8] The trend toward techniques that remove smaller parts of the cervix can be attributed to the fact that many of these are easy to do, they are of low cost and can be performed in an outpatient setting. Increased awareness of the impact of the more radical or deeper techniques on the risk of prematurity may have also contributed.[9–20]

The impact of different techniques on the risk of preinvasive and/or invasive recurrence remains therefore unclear. With some advocating the minimum radicality of treatment to prevent treatment-induced reproductive morbidity,[10 21] and others raising concerns about the increase in the risk of future invasion,[7 8] a definite answer regarding the relative merits and risks among the various treatment strategies is required.

Traditionally, treatment strategies are evaluated via large, expensive trials. Given the possibly comparable (and high) efficacy of most interventions for CIN, it is unlikely that any adequately powered RCT assessing the relative efficacy of different treatment techniques will ever be conducted. Such a trial would require thousands of women to reach the statistical power needed to detect differences in the preinvasive and invasive recurrence rates. In summary, there is currently a lack of adequately powered randomised evidence to allow us to compare the various interventions. However, there is a plethora of available observational studies in the field. These studies are a potentially valuable source of evidence and may act as a complement to the available randomised evidence, allowing us to more accurately assess the comparative effectiveness and safety of the various treatment alternatives. In this paper, we aim to perform a systematic review of both randomised and observational studies in the field and quantitatively synthesise their findings in meta-analyses.

Systematic reviews and pairwise meta-analyses are widely accepted as a useful tool in comparative effectiveness research and are commonly used to summarise, critically appraise and synthesise evidence from multiple studies. Investigators aiming to address a research question identify all relevant studies, evaluate their quality, synthesise their findings (meta-analysis) and interpret the provided evidence. Systematic reviews and meta-analyses have played a key role in providing evidence on the efficacy and safety of treatment methods and management strategies in cervical cancer prevention. However, the increased number of management strategies and multiple treatment options requires the use of more advanced evidence-synthesis methods.

Network meta-analysis (NMA) is an extension of pairwise meta-analysis, for the case when multiple treatments are available for the same condition. NMA has been recognised by the National Institute for Health and Care Excellence[22] and several international health technology assessment agencies[23 24] as a methodological tool that has the potential to increase precision in treatment effect estimates but also to infer on the clinical efficacy/safety between treatments that have never been compared in trials. NMA uses both direct evidence (ie, coming from studies comparing head-to-head the treatments of interest) and indirect evidence (ie, coming from studies comparing the treatments of interest via an intermediate common comparator),[25–28] allows the estimation of relative treatment effects between all available interventions and provides a clinically useful ranking of the different competing treatments. The methodology of NMA has never been used before to assess the comparative efficacy and complications of different treatment techniques used in the management of CIN. Furthermore, novel NMA methodologies will be employed to allow the use of both randomised and observational data.

The aim of this systematic review and NMA is to compare and clinically rank the alternative treatment techniques for CIN based on their efficacy, complications and adverse effects. This NMA forms part of the CIRCLE project (**C**ervical Cancer Incidence, **CIN** Recurrence and **R**eproduction after **L**ocal **E**xcision), which aims to generate a clinically useful ranking of alternative options for the treatment of CIN according to their efficacy (risk of preinvasive and invasive recurrence), morbidity and cost-effectiveness.

## METHODS AND ANALYSIS

This protocol is written in accordance with the Preferred Reporting Items for Systematic Review and Meta-Analysis Protocols (see online supplementary file 1).[29] PROSPERO registration will be updated if we make any amendments to this protocol. The start date was 1 October 2018 with expected end date on 1 October 2020.

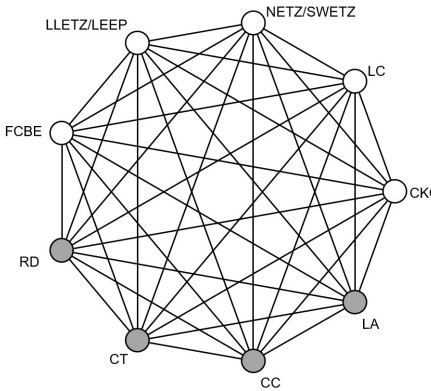

**Figure 1** Network of possible pairwise comparisons between eligible treatment methods. CC, cold coagulation; CKC, cold knife conisation; CT, cryotherapy; FCBE, Fischer cone biopsy excision; LA, laser ablation; LC, laser conisation; LEEP, loop electrosurgical excisional procedure; LLETZ, large loop excision of the transformation zone; NETZ, needle excision of the transformation; RD, radical point diathermy; SWETZ, straight-wire excision of the transformation zone.

## Eligibility criteria of studies

### Types of participants

We will include women of all ages treated with local surgical treatment for CIN (or glandular intraepithelial neoplasia) or microinvasive early cervical cancer (stage 1a1). We will only include women with histological diagnosis of CIN on punch biopsy or cone.

### Types of interventions

The treatment techniques for CIN are divided into excisional and ablative. The excisional include cold knife conisation, laser conisation, needle excision of the transformation zone, also known as straight-wire excision of the transformation zone, LLETZ, also known as loop electrosurgical excisional procedure and Fischer cone biopsy excisor, while the ablative include radical point diathermy, cryotherapy, cold coagulation and laser ablation. Figure 1 displays a network example of comparisons between studied treatment techniques.

### Outcome measures

*Primary outcome*

► Treatment failure rates defined as any abnormal cytology (atypical squamous cells of undetermined significance or worse) or histology (CIN1 or worse).

*Secondary outcomes*

► Treatment failure rates defined as high-grade abnormal cytology (high-grade squamous intraepithelial lesion or worse) or histology (CIN2 or worse).
► Treatment failure rates defined as residual or recurrent histologically proven CIN1 or worse.
► Treatment failure rates defined as residual or recurrent histologically proven CIN2 or worse.
► Human papillomavirus (HPV) positivity rates.

► Involved margins rates (incomplete excision of the lesion): both endocervical and ectocervical involvement.
► Perioperative or postoperative bleeding.
► Cervical stenosis.

Primary and secondary outcomes were chosen by clinical experts of the team. Treatment failure rates and HPV positivity rates will be reported at intervals of 6 to 12 months based on the available data and reported intervals in the included studies.

### Types of studies

We will include RCTs, quasi-RCTs and observational cohort studies comparing rates of treatment failure (recurrent/residual disease) or complications among the abovementioned surgical techniques. Single-arm studies not presenting a comparison will be excluded. Studies will be considered regardless of time or language.

### Information sources and search strategy

The Cochrane Gynaecological Cancer Specialised Register, Cochrane Central Register of Controlled Trials (CENTRAL), MEDLINE and EMBASE will be searched for eligible studies by an experienced librarian, as presented in online supplementary file 2. Metaregister, Physicians Data Query, www.controlled-trials.com/rct, www.clinicaltrials.gov, www.cancer.gov/clinicaltrials and WHO Registry Network (https://www.who.int/ictrp/network/en/) will be searched for ongoing studies. ZETOC (http://zetoc.mimas.ac.uk) and WorldCat Dissertations will be searched for conference proceedings and theses, respectively. References of the retrieved articles and meta-analyses will be hand-searched, the 'related articles' feature in MEDLINE will be employed and experts in the field will be contacted in an attempt to identify further reports of studies. Corresponding authors will be contacted for any relevant ongoing trials and unpublished data.

We will include both published and unpublished data and there will be no time, place or language restriction; articles in language other than English will be translated using online translation services.

### Study selection

Two team members will independently screen titles and abstracts of citations at level 1, using the reference management software Zotero. At level 2, the full text of all potentially eligible articles will be assessed using the same inclusion criteria. Disagreements will be resolved through discussion with a third review author.

### Data collection

Data from the included studies will be abstracted at level 3 by two reviewers independently using an a priori developed data collection form in Excel. The following data will be abstracted from the included studies: study characteristics, including author, publication year, country, study design, inclusion/exclusion criteria and intervention details; participant characteristics, including

age, CIN grade and smoking, and dropout rates; and outcome characteristics. In RCTs, we will prefer arm-level data (number of events and sample size per intervention arm for dichotomous data, and mean and SD per intervention arm for continuous data), but if these are missing, the study-level data will be used in the analysis, for example, reported ORs for dichotomous outcomes and mean differences (MDs) for continuous outcomes, along with a measure of uncertainty (eg, CI). For continuous outcome data not reported as means and SD, we will first contact the corresponding study authors for further information, but if no additional data are provided, we will perform imputation methods to derive approximate effect measures.[30 31] When an eligible study is observational, we will prefer adjusted treatment effect estimates accounting for the impact of potential confounders, but if these are missing, the unadjusted estimated treatment effects will be abstracted with a corresponding uncertainty measure (eg, CI). Disagreements will be resolved through consensus or the involvement of a third reviewer.

## Risk of bias assessment

RCTs will be assessed for quality and risk of bias using the Cochrane risk of bias tool[32] in the following domains: randomisation process, deviations from the intended interventions, missing outcome data, measurement of the outcome and selection of the reported result. The risk of bias in each domain, as well as the overall risk of bias, will be rated as 'low risk', 'some concerns' or 'high risk' after answering the signalling questions of each domain with 'Yes', 'Potentially Yes', 'Potentially No' or 'No'. Non-randomised studies (NRS) will be assessed using the ROBINS-I (Risk Of Bias In Non-Randomised Studies of Interventions) tool[33] with potential confounding factors: grade of treated CIN, age and smoking. The following domains will be assessed for NRS: confounding, selection of participants into the study, classification of interventions, deviations from intended interventions, missing data, measurement of outcomes and selection of the reported results. The risk of bias in each domain, as well as the overall risk of bias, will be rated as 'low', 'moderate', 'serious' or 'critical', after answering the signalling questions of each domain with 'Yes', 'Potentially Yes', 'Potentially No' or 'No'. Pairs of team members will independently assess the methodological quality and risk of bias of the eligible studies. Conflicts will be resolved through discussion or with a third investigator. When inadequate information is available from the studies to rate a risk of bias item, we will contact the corresponding study authors for clarification.

## Statistical synthesis
### Characteristics of included studies and network

For each outcome, we will produce a network plot (see, eg, figure 1) of the available evidence, as well as descriptive statistics, including comparison type, publication year, study design, outcome data and potential effect modifiers (eg, age).

## Pairwise meta-analyses

A random-effects meta-analysis will be conducted for each pairwise comparison in each outcome using the inverse variance model and the Hartung-Knapp-Sidik-Jonkman method to estimate each summary treatment effect and its 95% CI[34–36]. The between-study variance will be estimated with the restricted maximum likelihood estimator, whereas its 95% CI with the Q-profile approach.[34 37 38] We will also use the $I^2$ statistic along a 95% CI[39 40] to evaluate between-study heterogeneity. For continuous outcomes we will report the summary MDs, whereas for dichotomous outcomes we will use the summary ORs, along with a 95% CI. The *metafor* package[41] in R[42] will be used for all meta-analyses.

## Network meta-analyses
### Data synthesis

A random-effects NMA will be conducted, since we anticipate methodological and clinical between-study heterogeneity. We will assume common between-study variance parameter across treatment comparisons in the network, so that comparisons informed by a single study can borrow strength from the remaining network.[43 44] This assumption is clinically reasonable because all treatments included in the network of trials are of the same nature. The between-study variance will be estimated with the DerSimonian and Laird method of moments approach.[45] We will employ NMA models that account for different propensity of bias across different study designs as described in Efthimiou *et al*.[46] We will explore the impact of assigning different levels of credibility and subsequently downweight the NRS according to experts' opinion and the results of the ROBINS-I tool in several sensitivity analyses.

Similar to the pairwise meta-analysis, for continuous outcomes we will report the estimated MDs, whereas for dichotomous outcomes we will use the estimated ORs, with a 95% CI. Along the 95% CI for the summary effect size, we will report 95% prediction intervals, that is, the intervals within which the true underlying treatment effect is expected to lie in a new trial.[47] To rank the efficacy for each intervention, we will calculate the ranking probabilities for all treatments, the surface under the cumulative ranking curve (SUCRA) or P-scores and the mean ranks.[48 49] A rank-heat plot will be used to depict the SUCRA values or P-scores across all outcomes.[50] We will apply all NMA models in R[42] using the *netmeta* package[51] and *rjags*[52] package.

### Assessment of the transitivity assumption

One of the prerequisite assumptions in NMA is the transitivity assumption, under which the effect modifiers have a similar distribution across treatment comparisons in a network.[27 53 54] For the participants' characteristics that are described in the inclusion criteria of our systematic review (section type of participants), it is reasonable to assume that all treatments we plan to compare (section type of interventions) are 'jointly randomisable'. That means that any patient that fulfils that inclusion criteria could potentially be assigned to any of the interventions. Potential effect modifiers expected to influence the estimated treatment

effects include year of study, level of income of study country (as defined by World Bank[55]), method of ascertainment of exposure/outcome (hospital records, registries or interviews/questionnaires), age, smoking and grade of CIN. For each pairwise comparison with available direct evidence, we will summarise these characteristics and will visually inspect the similarity of the identified studies. We will also investigate the inclusion and exclusion criteria of all studies, to make sure that patients, treatments and outcomes in the studies are sufficiently similar in all aspects that might modify relative treatment effects. More specifically, we will compare the patient characteristics (such as severity, age, parity, etc) across the different treatments. If these characteristics are found to have a similar distribution across treatments, then transitivity is supported. If differences are found, then these will be addressed in subgroup and sensitivity analyses.

### Assessment of statistical inconsistency

Consistency in a network of trials will be evaluated both locally and globally. We will first assess the consistency assumption locally by separating the direct from the indirect evidence for every comparison in a network to make judgements about their statistical differences, using the back-calculation method.[56] Then we will assess consistency in each network globally using the design-by-treatment interaction model.[57] We will conceptually explore for potential intransitivity in every network even in the absence of evidence for inconsistency, since the inconsistency tests have low power to detect true inconsistency.[58 59] If no substantial inconsistency is identified in the network of RCTs, we will then evaluate the agreement between RCTs and NRS using the same local and global approaches. Both local (back-calculation method) and global (design-by-treatment interaction model) assessments will be performed under the random-effects model in R[42] using the *netmeta* package.[51]

In the NMA including both RCTs and NRS, we will assess for differences between the different study designs.[46] For each treatment comparison we will summarise evidence by up to four different types: direct randomised, indirect randomised, direct non-randomised and indirect non-randomised. If important discrepancies between these types are found, these will be investigated to confirm that the transitivity assumption holds (eg, when randomised and non-randomised evidence are very different in terms of populations, interventions, and so on, the transitivity assumption may be violated). If disagreement occurs for a certain characteristic, this will be explored through a network meta-regression model.[26]

### Exploring heterogeneity and inconsistency: subgroup analyses and meta-regression

The between-study heterogeneity will be explored by comparing the estimated between-study variance with the empirical distribution derived by Rhodes *et al* for continuous data[60] and the one derived by Turner *et al* for dichotomous data.[61] We will also compare 95% CIs with the 95% prediction intervals to infer on the magnitude of the between-study variance.

If at least 10 studies are available, the following potential sources of heterogeneity and/or inconsistency will be explored for the primary outcome using subgroup or metaregression analyses: year of study, level of income of study country (as defined by World Bank,[55] method of ascertainment of exposure/outcome (hospital records, registries or interviews/questionnaires), age, smoking, grade of CIN and disease severity (eg, women treated for high-grade CIN, exclusion of cases of microinvasion). To minimise potential bias due to confounding from NRS (eg, type of treatment or outcome affected by severity), we will also perform a sensitivity analysis excluding NRS without adjusted effect estimates.

### Reporting bias and small study effects

We will assess small-study effects by visually exploring the funnel plot for each treatment, and the comparison-adjusted funnel plot[62] when at least 10 studies are available. We will also conduct a network meta-regression using the study variance as a covariate.[63 64]

### Assessment of the credibility of the evidence

For the primary outcome, two team members will determine the degree of confidence in the estimated NMA results using CINeMA (Confidence In Network Meta-Analysis)[65] and the relevant online tool (http://cinema.ispm. ch/). The six CINeMA domains, within-study bias (ie, risk of bias in the included studies), across-study bias (ie, publication and reporting bias), indirectness, imprecision, heterogeneity and incoherence (ie, differences between direct and indirect evidence),[65] will first be rated as high quality and then they will be downgraded if judged appropriate to moderate, low or very low quality.

## PATIENT AND PUBLIC INVOLVEMENT

Patients and the wider public have been involved from the design of this proposal through clinics and the Jo's Cervical Cancer Trust. They have assisted study design and to formulate the research questions. Their involvement will continue throughout the study on regular 6 monthly meetings and will guide the priority questions to be addressed, the development of research reports in lay language and the dissemination of the results.

## ETHICS AND DISSEMINATION

We do not require ethical approval for this review. We aim to disseminate the results to clinicians, academic researchers, health agencies, decision-makers, patients and the public. We will publish the results in high impact open access journals and disseminate findings through presentations at medical conferences. The data will become available in public repositories. We will develop information sheets and briefings, highlighting the key findings and circulate newsletters. We will work

closed with the Jo's Trust, charity in cervical cancer that frequently organises events to educate the public and also engage the media with interviews. We circulate findings in the Imperial College London web page and will circulate newsletters.

**Author affiliations**
[1]Department of Surgery and Cancer, Faculty of Medicine, Institute of Reproductive and Developmental Biology, Imperial College London, London, UK
[2]Imperial College Healthcare NHS Trust, London, UK
[3]Department of Primary Education, School of Education, Panepistimio Ioanninon, Ioannina, Greece
[4]Institute of Social and Preventive Medicine (ISPM), University of Bern, Bern, Switzerland
[5]Department of Obstetrics and Gynaecology, University of Helsinki and Helsinki University Hospital, Helsinki, Finland
[6]Department of Health Policy, London School of Economics and Political Science, London, UK
[7]Department of Gynaecologic Oncology, Lancashire Teaching Hospitals NHS Foundation Trust, Preston, UK
[8]Department of Obstetrics and Gynaecology, University of Ioannina and University Hospital of Ioannina, Ioannina, Greece

**Acknowledgements** We thank patients and patient representatives for their contribution to designing this study and developing the research questions.

**Contributors** The study was conceived and designed by MK, GS and EP. The protocol was drafted by AA, MK, AAV, OE, IK, GS and was revised critically for important intellectual content by all authors (AA, AAV, OE, IK, HN, SB, MP, PM-H, PB, EP, GS, MK).

**Funding** This work is supported by National Institute for Health Research (NIHR) Research for Patient Benefits (Grant Reference Number PB-PG-0816-20004, P67307). It is also supported by the British Society of Colposcopy Cervical Pathology Jordan/Singer Award (P47773) and the Imperial College Healthcare Charity (P47907) (MK). AAV is funded by the European Union's Horizon 2020 (No 754936). None of the funders have any influence on the study design; in the collection, analysis and interpretation of data; in the writing of the report; and in the decision to submit the article for publication.

**Competing interests** MK has received travel and conference expenses, honoraria and consultancy fees for commercial companies (Inovio, MSD, etc); these activities are not related to the project.

**Patient consent for publication** Not required.

**Provenance and peer review** Not commissioned; externally peer reviewed.

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
