## [Reviewer comments · BMJ Open]

ARTICLE DETAILS

TITLE (PROVISIONAL)	Comparative efficacy and complication rates after local treatment for cervical intra-epithelial neoplasia and stage 1a1 cervical cancer: protocol for a systematic review and network meta-analysis from the CIRCLE Group
AUTHORS	Athanasiou, Antonios; Veroniki, Areti Angeliki; Efthimiou, Orestis; Kalliala, Ilkka; Naci, Huseyin; Lever, Sarah; Paraskevaidi, Maria; Martin-Hirsch, Pierre; Bennett, Philip; Paraskevaidis, Evangelos; Salanti, Georgia; Kyrgiou, Maria

VERSION 1 - REVIEW

REVIEWER	Jose Guilherme Cecatti The University of Campinas Brazil
REVIEW RETURNED	08-Feb-2019

GENERAL COMMENTS	This is a good and exceptionally well written protocol for a systematic review focusing efficacy and complication rates after treatment for CIN. The proposal is well developed and goes deeper in the proposed analysis and I strongly recommend its publication. I have only three minor suggestions to add: Page 5, line 58: it states the review will not be restricted by language. Did the authors really have alternative procedures for dealing with studies published in other languages? For instance in Chinese, Arabian, Portuguese, Thai, Russian, etc? If yes, please clearly state taht. Page 6, lines 9-10: besides the registries already stated for searching ongoing trials, I would also recommend to check all international registries of trials, as recognized by the WHO. Page 6, data collection, lines 34-35: I would also recommend to collect the information on from what country the study comes from. This could also be used as a factor for stratification of the analysis to see if the level of income country or HDI influence the results.
---

REVIEWER	h.j.van beekhuizen Erasmus MC Rotterdam The Netherlands
REVIEW RETURNED	08-Mar-2019

GENERAL COMMENTS	A more recent, large, population-based study from Sweden ⁷ : this study is from 2014, I would not call this recent.
--

REVIEWER	Maaïke Bleeker Dept. Pathology, Amsterdam UMC, VU University Medical Center, Amsterdam, The Netherlands
REVIEW RETURNED	12-Apr-2019

GENERAL COMMENTS	It will be useful to use novel analysis techniques to retrieve good information on the outcome after treatment. A meta-analysis is therefore useful. Comments: 1) It should be emphasized that the outcome of the different methods is difficult to compare since the choice of treatment is not chosen at random and is influenced by the underlying lesion as well as the age and pregnancy wishes of the women to be treated and the setting of the clinical care system. The choice of treatment for each patient will be a different 'cost-benefit analysis/balance' in which the pros and cons will be discussed and weighed up but this information is not included in the analysis. Thus, for instance it is difficult to compare the outcome of CIN2+ in women who underwent a cone because of a micro-invasive carcinoma with those women who underwent ablative therapy for a low grade CIN lesion (note: ablative therapy will not be a regular treatment option when micro-invasive disease is present). I cannot see clearly how the meta-analysis can overcome this issue and produce clinical ranking of treatments. In other words, can NMA overcome the issue that the type of treatment choice is related to lesion type? If so, this should be clearly stated. If not, this method cannot produce clinical ranking of treatments. 2) It would be useful to look at outcome per treatment type. I think that the primary outcome measure should be recurrent disease (so when a woman is treated for low grade CIN, it may be good to look at recurrent low grade CIN). Although low grade CIN is not an indication for treatment, many treatments occur for low grade CIN. How is dealt with the variety of disease type and load in correlation with outcome? I think that a histological-proven CIN1 is a difficult outcome measure after treatment for a CIN2+ lesion. 3) The secondary outcome measure of involved margins rates should be clarified. Do you mean the involved sites? 4) It would be useful to include prematurity as a secondary outcome measure as well 5) The time after treatment to measure outcomes should be included
---

VERSION 1 – AUTHOR RESPONSE

Reviewer(s)' Comments to Author:

Reviewer 1: Jose Guilherme Cecatti

Institution and Country: The University of Campinas, Brazil

This is a good and exceptionally well written protocol for a systematic review focusing efficacy and complication rates after treatment for CIN. The proposal is well developed and goes deeper in the proposed analysis and I strongly recommend its publication.

We thank the reviewer for his supporting comments.

I have only three minor suggestions to add:

Page 5, line 58: it states the review will not be restricted by language. Did the authors really have alternative procedures for dealing with studies published in other languages? For instance in Chinese, Arabian, Portuguese, Thai, Russian, etc? If yes, please clearly state taht.

We have now added clarification for this in the section 'Information sources and search strategy':

'articles in language other than English will be translated using online translation services.'

Page 6, lines 9-10: besides the registries already stated for searching ongoing trials, I would also recommend to check all international registries of trials, as recognized by the WHO.

Thank you. We have now added this source in the section 'Information sources and search strategy'.

'Metaregister, Physicians Data Query, www.controlled-trials.com/rct, www.clinicaltrials.gov, www.cancer.gov/clinicaltrials and WHO Registry Network (<https://www.who.int/ictrp/network/en/>) will be searched for ongoing studies.'

Page 6, data collection, lines 34-35: I would also recommend to collect the information on from what country the study comes from. This could also be used as a factor for stratification of the analysis to see if the level of income country or HDI influence the results.

Thank you. It was our intention to collect data on the country. We have missed this in the text and the word "country" has now been added in section 'Data Collection'

We will also perform subgroups analyses according to the level of income of the country as per reviewer's recommendation. This has now been added in the section 'Exploring heterogeneity and inconsistency: subgroup analyses and meta-regression.'

'If at least 10 studies are available, the following potential sources of heterogeneity and/or inconsistency will be explored for the primary outcome using subgroup or meta-regression analyses: year of study, level of income of study country (as defined by World Bank), method of ascertainment of exposure/outcome (hospital records, registries or interviews/questionnaires), age, smoking and grade of CIN.'

It has also been added in the section Assessment of the transitivity assumption: 'Potential effect modifiers expected to influence the estimated treatment effects include year of study, level of income

of study country (as defined by World Bank), method of ascertainment of exposure/outcome (hospital records, registries or interviews/questionnaires), age, smoking and grade of CIN.'

Reviewer 2: h.j.van beekhuizen

Institution and Country: Erasmus MC Rotterdam The Netherlands

A more recent, large, population-based study from Sweden⁷: this study is from 2014, I would not call this recent.

Thank you for the comment. We have deleted 'more recent'.

Reviewer 3: Maaïke Bleeker

Institution and Country: Dept. Pathology, Amsterdam UMC, VU University Medical Center, Amsterdam, The Netherlands

It will be useful to use novel analysis techniques to retrieve good information on the outcome after treatment. A meta-analysis is therefore useful.

We thank the reviewer for this supporting comment.

Comments:

1) It should be emphasized that the outcome of the different methods is difficult to compare since the choice of treatment is not chosen at random and is influenced by the underlying lesion as well as the age and pregnancy wishes of the women to be treated and the setting of the clinical care system. The choice of treatment for each patient will be a different 'cost-benefit analysis/balance' in which the pros and cons will be discussed and weighed up but this information is not included in the analysis. Thus, for instance it is difficult to compare the outcome of CIN2+ in women who underwent a cone because of a micro-invasive carcinoma with those women who underwent ablative therapy for a low grade CIN lesion (note: ablative therapy will not be a regular treatment option when micro-invasive disease is present).

I cannot see clearly how the meta-analysis can overcome this issue and produce clinical ranking of treatments. In other words, can NMA overcome the issue that the type of treatment choice is related to lesion type? If so, this should be clearly stated. If not, this method cannot produce clinical ranking of treatments.

Thank you for your valuable comment.

The bias possibly introduced by treatment selection is present at two levels; within studies and across studies.

The first (within studies) is associated with the fact that non randomised trials are included and will be addressed using the classic approaches; estimating the Risk of Bias using ROBINS-I and by extracting as far as possible confounder-adjusted effect sizes (adjusted for variables that could have suggested the choice of the treatment within the study, e.g. CIN).

The second is associated with the comparability of the studies examining different treatments and the NMA assumption of transitivity and consistency. We will approach this at two levels; conceptually and epidemiologically.

- First, we state now in our protocol (see section Network meta-analyses, Assessment of transitivity assumption) that “For the participants characteristics that are described in the inclusion criteria of our systematic review (section Eligibility criteria, Type of participants), it is reasonable to assume that all treatments we plan to compare (listed in section Eligibility criteria, Type of interventions) are “jointly randomisable”. That means that any patient that fulfils that inclusion criteria, could potentially be assigned to any of the interventions.”.

- The second way to evaluate the transitivity assumption is by comparing the study characteristics across treatment comparisons. We discuss this in section Network meta-analyses, Assessment of transitivity assumption. We explain that “More specifically we will compare the patient characteristics (such as severity, age, parity etc) across the different treatments. If these characteristics are found to have a similar distribution across the treatments then transitivity is supported. If differences are found, then these will be addressed in subgroup and sensitivity analyses.”.

We will also perform sensitivity analysis including only women with high-grade CIN and sensitivity analyses excluding studies or subgroup of patients when possible with microinvasive cancer and by excluding studies without adjusted effect estimates, as described in section Exploring heterogeneity and inconsistency: subgroup analyses and meta-regression:

‘If at least 10 studies are available, the following potential sources of heterogeneity and/or inconsistency will be explored for the primary outcome using subgroup or meta-regression analyses: year of study, level of income of study country (as defined by World Bank⁵⁵), method of ascertainment of exposure/outcome (hospital records, registries or interviews/questionnaires), age, smoking, grade of CIN, and disease severity (e.g., women treated for high-grade CIN, exclusion of cases of microinvasion). In order to minimise potential bias due to confounding from NRS (e.g. type of treatment or outcome affected by severity), we will also perform a sensitivity analysis excluding NRS without adjusted effect estimates.’

2) It would be useful to look at outcome per treatment type. I think that the primary outcome measure should be recurrent disease (so when a women is treated for low grade CIN, it may be good to look at recurrent low grade CIN). Although low grade CIN is not an indication for treatment, many treatments occur for low grade CIN. How is dealt with the variety of disease type and load in correlation with outcome?

I think that a histological-proven CIN1 is a difficult outcome measure after treatment for a CIN2+ lesion.

Thank you for this comment. We agree that in principle residual/recurrent disease of any grade as the primary outcome would allow us to capture also women treated with low-grade disease. We also agree that a histologically-confirmed CIN1 outcome might be difficult to measure as biopsies are frequently not taken and we have therefore included both cytological and histological abnormalities in our primary outcome.

The changes in the outcomes are shown in page 5 and also copied below:

'Outcome measures

Primary outcome

- Treatment failure rates defined as any abnormal cytology [ASCUS (atypical squamous cells of undetermined significance) or worse] or histology (CIN1 or worse)

Secondary outcomes

- Treatment failure rates defined as high-grade abnormal cytology (HSIL (high-grade squamous intraepithelial lesion) or worse) or histology (CIN2 or worse)
- Treatment failure rates defined as residual or recurrent histologically-proven CIN1 or worse
- Treatment failure rates defined as residual or recurrent histologically-proven CIN2 or worse
- HPV positivity rates
- Involved margins rates (incomplete excision of the lesion): both, endocervical, ectocervical involvement
- Peri-operative or post-operative bleeding
- Cervical stenosis

3) The secondary outcome measure of involved margins rates should be clarified. Do you mean the involved sites?

Thank you. By involved margins we refer to the rate of incomplete excision of the lesion as reported by the histopathologist. This refers to excisional techniques alone and can affect the endocervical, ectocervical margins or both margins. Incomplete excision is an important determinant of recurrent or persistent CIN-lesion during follow-up (Arbyn Lancet Oncol 2017). We wish to assess this outcome as an indirect outcome of risk of recurrence.

We have amended the text for clarity:

'...Involved margins rates (incomplete excision of the lesion): both, endocervical, ectocervical involvement'

4) It would be useful to include prematurity as a secondary outcome measure as well

We agree this is an important outcome but prematurity as well as other reproductive outcomes are addressed in a different protocol led by the same research group.

5) The time after treatment to measure outcomes should be included

We agree and have now included timepoints in the section 'Outcome measures' for the time-dependent outcome measures of interest:

'Treatment failure rates and HPV positivity rates will be reported at 6 to 12 months intervals based on the available data and reported intervals in the included studies.'

VERSION 2 – REVIEW

REVIEWER	Maaïke Bleeker Amsterdam UMC, VUmc, Amsterdam, The Netherlands
REVIEW RETURNED	24-Jun-2019

GENERAL COMMENTS	I am satisfied with the authors answers to my earlier comments
--